# Optimizing tuberculosis screening for immigrants in southern New Brunswick: A pilot study protocol

Isdore Chola Shamputa[1]*, Duyen Thi Kim Nguyen[2,3], Doaa Higazy[4], Amani Abdelhadi[5], Hope MacKenzie[6], Michelle Reddin[2], Kimberley Barker[2], Duncan Webster[6,7,8]

1 Department of Nursing & Health Sciences, University of New Brunswick, Saint John, New Brunswick, Canada, 2 Ministry of Health, Government of New Brunswick, Saint John, New Brunswick, Canada, 3 Faculty of Business, University of New Brunswick, Saint John, New Brunswick, Canada, 4 Saint John Newcomers Centre Inc., Newcomer Settlement, Saint John, New Brunswick, Canada, 5 Newcomer Connection, The YMCA of Greater Saint John, Saint John, New Brunswick, Canada, 6 Division of Microbiology, Department of Laboratory Medicine, Saint John Regional Hospital, Saint John, New Brunswick, Canada, 7 Faculty of Medicine, Dalhousie University New Brunswick, Saint John, New Brunswick, Canada, 8 Division of Infectious Diseases, Department of Medicine, Saint John Regional Hospital, Saint John, New Brunswick, Canada

* chola.shamputa@unb.ca

**Data Availability Statement:** No datasets were generated or analyzed during the current study. All relevant deidentified data will be made publicly

## Abstract

### Introduction

Immigrants from high tuberculosis-burdened countries have been shown to have an increased risk of latent tuberculosis infection (LTBI). To reduce the risk of increased tuberculosis cases in Canada, the country has a comprehensive immigration medical examination process that identifies individuals with active tuberculosis using chest X-ray; however, it fails to identify LTBI. The lack of LTBI identification is concerning because immigrants with LTBI are at an increased risk of developing active tuberculosis within their first few years of migration due to stressful experiences common to many immigrants.

### Objectives

The goal of this pilot study is to improve the current LTBI screening protocols among immigrants from high tuberculosis incidence countries and to better prevent and manage tuberculosis cases, by introducing an LTBI screening pilot program. The objectives are threefold: 1) to screen LTBI in immigrants from high tuberculosis incidence countries, including immigrants identified as being at risk of LTBI by the NB health care system, using the QuantiFERON-TB Gold Plus interferon-gamma release assay (IGRA); 2) to offer LTBI treatment and supports to those identified as having LTBI; and 3) to assess immigrant and health care providers (HCPs) satisfaction of the LTBI screening pilot program.

### Methods

This cross-sectional study seeks to recruit 288 participants. Participants will be recruited via posters, social media platforms, invitations at immigrant wellness check-ups, presentations to local ethnocultural groups, and by snowball sampling. Consenting participants will be

available when the study is completed and
published.

**Funding:** This project was supported by a grant
from the Chesley Family Research Award (https://
medicine.dal.ca/) and the New Brunswick Health
Research Foundation (https://nbhrf.com/en/) (ICS,
DTKN, DH, AA, HM, KB, DW). The study also
received funding from the New Brunswick
Innovation Fund (Emerging projects ref #:
EP_2022_017) (https://nbif.ca) (ICS, DTKN, DH,
AA, HM, KB, DW), the University of New Brunswick
(Student Work Program) (https://www.unb.ca)
(ICS), New Brunswick Health Research Fund (ICS)
and in-kind donation of interferon gamma release
assay (QuantiFERON®-TB Gold Plus) tests from
Qiagen Inc (https://www.qiagen.com/us) (ICS, KB,
DW). The funders had and will not have a role in
study design, data collection and analysis, decision
to publish, or preparation of the manuscript.

**Competing interests:** I have read the journal's
policy and the authors of this manuscript have the
following competing interests: [I (Isdore Chola
Shamputa) currently serve as an Academic Editor
for PLOS Global Public Health]. All the other
authors have declared that no competing interests.

asked to submit a blood sample for LTBI screening; if positive, participants will be assessed and offered treatment for LTBI based on clinical assessment. Participants and HCPs' feedback will be gathered via short questionnaires. For the quantitative portion of the study, descriptive statistics will be used to summarize participant characteristics and feedback. Simultaneous logistic regression will be performed to identify variables associated with the IGRA test outcome and evidence of increased CD8 T-cell immune response among those found to be LTBI-positive. Qualitative results will be analyzed using inductive thematic analysis.

## Discussion

The findings from this study will allow us to understand the role of the IGRA LTBI screening assay and its feasibility and acceptability by immigrants and HCPs in New Brunswick. The findings will additionally provide information on the enhancers and barriers of LTBI screening and management useful in determining how best to expand the LTBI screening program if deemed appropriate.

## Introduction

Tuberculosis (TB) is a major public health problem worldwide. It is estimated that a quarter of the planet's population is infected with *Mycobacterium tuberculosis*, the bacteria that causes TB. In 2019, an estimated 10 million people suffered from active TB, and 1.4 million died from the disease [1]. The majority of people suffering from TB reside in low- and middle-income countries [1]. However, due to increased immigration and international travel, there is a growing number of TB cases being reported in high-income countries, with immigrants showing an increased risk of having latent TB infection (LTBI) and developing active TB within the first five years of migration [2]. Stressful living conditions, such as food insecurity, low-language fluency, cultural barriers, and housing insecurity, disproportionately affect recent immigrants and increase the likelihood of activating TB among LTBI-positive immigrants post-migration [3].

About 71% of TB cases in Canada are among non-Canadian-born individuals from high-TB incidence countries. In New Brunswick (NB), the number of incoming immigrants has increased sixfold in the last two decades [4]. NB has recently increased its intake of immigrants even more significantly, with the introduction of the Atlantic Immigration Program, which welcomed 16,000 immigrants to NB in the last three years alone [5]. Through a population growth strategy, NB aims to accept 7,500 more immigrants per year until 2024, the maximum allowed by the federal government [6]. With the foreseeable increase in NB's immigrant population, many from high-burden TB countries, it is imperative that the province's health care system prepare to prevent and manage the potential increased incidence of easily transmissible diseases, such as TB. Strategies directed toward improved LTBI surveillance, integrated LTBI screening, and proper active TB management may decrease the rate of infectious TB cases and reduce transmission and secondary cases [7,8]. To the best of our knowledge, this study is unique in Atlantic Canada.

This study uses the QuantiFERON-TB Gold Plus (QFT-Plus) interferon-gamma release assay (IGRA) (QIAGEN), which elicits both CD4 and CD8 T-cell response as a screening assay for LTBI [9,10]. The IGRA is available in Saint John, NB; the only laboratory processing and analyzing IGRA's in Atlantic Canada. Compared to traditional LTBI screening tools, such

as the tuberculin skin test, the IGRA requires a single blood draw, is not affected by the bacille Calmette-Guerin vaccination or previous exposure to most non-tuberculous mycobacteria, and has fewer issues with inter-rater reliability [11,12]. Thus, the IGRA is an excellent means to screen for LTBI, as it 1) provides more accurate identification of LTBI cases, 2) thus helping to ensure the optimal management of patients, and 3) with associated programming may help reduce the transmission of TB by individuals progressing from LTBI to active TB.

## Statement of objectives

To improve the current TB screening protocols used in NB, and to better prevent and manage TB cases, our goal is to implement a LTBI screening pilot program in southern NB, using the IGRA as the primary screening method.

To achieve our goal, our **objectives** are threefold:

1. to screen LTBI in 288 immigrants from high TB incidence countries (as identified by ≥40/ 100 000 population) [2], including immigrants identified as being at risk of LTBI by the NB health care system, using the IGRA;

2. to offer LTBI treatment and supports to those identified as LTBI-positive; and

3. to assess newcomer and HCPs' satisfaction with the LTBI screening pilot program.

This study builds upon our previous research, which investigated the knowledge, attitudes, and beliefs of 43 newcomers in southern NB regarding LTBI. Our previous study identified several key findings: 1) Most of the participants were willing to get tested and treated for LTBI; 2) participants desired an increased awareness of LTBI; 3) participants emphasized the importance of supporting people with an LTBI diagnosis, and 4) participants shared a common experience of general stigma associated with TB [13].

## Materials and methods

Our study will employ a cross-sectional study design, and will be guided by the sex-gender-based analysis plus approach [14] and an equity, diversity, and inclusion lens [15].

### Participants

To ensure our study has adequate power to detect statistical significance (α level = 0.05), we aim to recruit 288 participants [16]. In an effort to be as inclusive as possible, four broad eligibility criteria will guide recruitment. Potential participants must be: 1) born outside of Canada; 2) 19-years of age or older at the time of consent; 3) came from, or resided in, a country with a TB incidence of ≥40/100 000 population [2] before arriving in Canada [17], are at high-risk of TB (e.g., having a TB-positive partner), or government assisted refugees who participate(d) in a Post Arrival Health Assessment by the NB Public Health and are (were) tested for LTBI; and 4) a resident in southern NB (i.e., Charlotte, Saint John, and Kings counties, extending from Sussex to St. Stephen). To conduct stratified analyses on several key factors, we aim to have a broad and balanced participant recruitment based on gender, sex, ethnicity, age, immigration stream, and year of arrival to Canada.

### Recruitment

**Materials.** We will examine the implementation and satisfaction of the LTBI screening pilot program using several data collection tools. All data collection tools for immigrants are

available in seven languages (English, French, Tagalog, Somali, Farsi/Dari, Mandarin, and Spanish) for enhanced comprehension (S1 and S2 Appendices).

1. *Participant demographics questionnaire and history*. We will collect key demographic information relevant to our understanding of LTBI and the screening process, such as sex and gender, country of origin, date of arrival to Canada, immigration stream of entry (S3 Appendix). We will also measure factors that may intersect with sex and gender, such as ethnicity, and age [18].

2. *Participant survey*. Participant feedback regarding their experience of the pilot screening program will be gathered through a series of 5-point Likert scales and open-ended questions (S2 Appendix). Participants have the option to complete the questionnaire in hard copy or online. Questions will relate to equity, diversity, and inclusion, such as barriers and enablers of LTBI testing, including language, cultural issues, accessibility of the blood collection facility, their interaction with the health care professionals, and ease/difficulty of participating in this study.

3. *HCP survey*. HCP demographic information and feedback will be gathered via a questionnaire, and feedback questions will address barriers and enablers of the pilot screening program, such as challenges encountered during their interactions with the participants and management and follow-up (S4 Appendix).

## Procedure

This study is being conducted in multiple steps, with some steps occurring concurrently.

**1) Developing and pilot testing questionnaires.** Participant questionnaires and health care questionnaires were pilot-tested with a sample of immigrants and HCPs, respectively, for clarity, coherence, and time of completion [19]. To help prevent respondent fatigue, we aimed to have the survey completed in under 10 minutes [20] and written at a grade 3 level for maximum comprehension [19].

**2) Research ethics board approval.** This study received ethics approval from; 1) the Horizon Health Network; and 2) the University of New Brunswick Research Ethics Boards (reference numbers: 033–2021 and 2021–3046, respectively).

**3) Recruitment.** Participants are being recruited through several mediums, including posters distributed by newcomer-serving organizations and their various social media platforms, invitations at newcomer wellness check-ups, through presentations to local ethnocultural groups, and snowball sampling. As indicated above, recruitment materials are available in seven of the most commonly spoken languages among immigrants in southern NB.

**4) LTBI screening.** Interested participants are screened virtually by a research team member to ensure they meet the eligibility criteria, are provided with study information, and understand the consent form. If the criterion is met and participants demonstrate a clear understanding of the informed consent form, then oral informed consent will be obtained by a research team member before participating in the study. Participants will receive an e-gift card of $20 after blood collection and another $20 e-gift card after completing the online participant survey for a total of $40 compensation for their time.

**5) Sample collection.** Blood samples will be collected by a certified phlebotomist at the Young Men Christian Association of Greater Saint John on dedicated days following existing coronavirus disease 2019 operational protocols. Alternate blood collection sites include the Saint John Regional Hospital, Saint Joseph's Hospital, the Uptown Community Clinic in Saint John, Charlotte County Hospital, the Fundy Health Clinic and Sussex Hospital.

**6) Laboratory analyses and tuberculosis testing results.** The IGRAs will be analyzed at the Saint John Regional Hospital's Microbiology Laboratory according to the recommended protocols [21]. IGRA results and associated diagnostic data will be accessed by the research team via electronic TB client registration systems at the Saint John Regional Hospital and the NB Public Health data system for all participants diagnosed with latent or active TB. IGRA results will be communicated to participants by the attending HCPs following existing Horizon Health Network procedures. Positive results will be assigned appropriate triaging for follow-up and treatment.

**7) LTBI follow-up for LTBI-negative participants** Individuals with a negative test are being contacted with their results by an HCP.

**8) LTBI follow-up and supports for LTBI-positive participants.** Individuals with positive test results will undergo a further clinical assessment to determine the need for additional testing to rule out an active TB diagnosis. Those with confirmed LTBI will be counselled, offered treatment, and uniquely managed for optimal care. Participants with confirmed active TB will be treated according to appropriate protocols [22]. All individuals will be provided counseling and written information with regards to TB/LTBI, and information pertaining to relevant medications. Support and follow-up will be provided during treatment to ensure adherence, and to address stigma- and treatment-related concerns.

**9) Participant and HCP surveys.** Participant questionnaires will be administered virtually by a research team member after i) the participant has received a negative IGRA result, ii) after the completion of treatment for those with a positive IGRA result, were assessed by an HCP, offered and agreed to be treated, or iii) after receiving a positive IGRA result, were assessed by an HCP, offered but declined treatment. Questionnaires will be administered to all HCPs once our study has reached its goal of 288 participants. The data will be linked to previous demographic information and test results using unique, de-identifiable codes.

## Data analysis plan

To help monitor and ensure we meet our recruitment goals, we will conduct data collection and data analyses concurrently.

## Statistical analyses

Underlying our analyses will be the sex-gender-based analysis plus (SGBA+) approach, and an equity, diversity, and inclusion lens. Descriptive statistics (i.e., frequencies, means, and ranges) will be computed to summarize participant characteristics and feedback. Simultaneous logistic regression will be performed to identify variables associated with the IGRA test outcome (positive or negative) and evidence of increased CD8 T-cell response among those found to be LTBI-positive. Predictors for the logistic regression will be age, sex, gender, country of origin, immigration stream, and the presence of co-morbidities. Power analysis for the logistic regression using the 10 events per predictor variable rule of thumb [$(10^* k)/p$, where k is the number of predictors (6 predictors), and p is the lowest event rate] and assuming a conservative 25% event rate for LTBI, indicates a minimum sample size of 240 participants is required to power the regression. To account for potential participant, drop-out, an additional 20% will be added to the sample size, leading to a need for 288 participants. An alpha of 0.05 will be used for the regression analysis. All statistical analyses will be performed using Stata statistical software (version 16; StataCorp, 2019) [23] and/or IMB SPSS Statistics software, version 27 (Armonk, NY) [24].

## Qualitative analyses

All qualitative feedback will be inputted into Microsoft Excel and Word for coding and thematic analyses as described elsewhere [25]. Briefly, two research team members will independently familiarize themselves with all of the data. Next, initial codes will be generated independently and collaboratively with a third-team member. Subsequently, codes will be integrated into emerging themes; these will be reviewed by the entire team and modified if necessary.

## Data security

Our data will be confidential (i.e., de-identified) and triple safeguarded: 1) data files will remain password protected; 2) laptops/Personal Computers will also be password-protected; and 3) laptops/Personal Computers will remain in a locked and secured possession of the users.

All electronic data will be stored on a secure University of New Brunswick Saint John drive that will only be accessed by members of the study team. In addition, the master record will be password protected for additional security. All paper data will be shredded, and computer files will be permanently deleted after 5 years to allow for the dissemination of study findings. Data will be reported in aggregate form. For the purposes of identifying interested individuals who wish to be contacted with the final report, it is necessary to maintain identifiers in the individual consent forms.

Participant recruitment for this study has started and will continue until we reach our targeted sample size.

## Discussion

This study seeks to improve the current LTBI screening protocols among immigrants from high TB incidence countries and to better prevent and manage TB cases in southern NB. At the completion of this study, it is anticipated that i) we will understand the role of the IGRA LTBI screening assay and its feasibility and acceptable by immigrants and HCPs, and ii) the findings will avail information on the enhancers and barriers of LTBI screening and management that will be useful in determining how best to expand the program province-wide if deemed appropriate.

Identifying and treating individuals with LTBI will reduce the pool of individuals with LTBI, thereby lowering their potential to develop active TB. In turn, this will potentially reduce health costs associated with the transmission, contact tracing, treatment of active TB, and follow-up of TB patients with the promotion of a healthier NB population. Further, the detection and treatment of individuals with LTBI have several other advantages; first, given that an estimated one-quarter of the world's human population is infected with TB [1], detecting and treating LTBI is one of the critical strategies for the elimination of TB as a global public health threat [26]. Second, mathematical modeling studies have demonstrated that through screening and control strategies targeting LTBI, the development of active TB can be reduced between 20.6 [27] and 40% on a population level [28]. Third, the prevention of active TB by detecting and treating individuals with LTBI can also help prevent numerous health problems and post-TB sequelae that may be experienced by individuals despite adequate and successful treatment of active TB [29]. In addition, the model that this research will develop may benefit the prevention and control of other diseases in NB. Thus, our research will likely significantly impact keeping Canadians, and New Brunswickers in particular, safe. Finally, the findings of this study will help lobby for the development of an LTBI program among immigrants from high TB-incidence countries in southern NB.

Knowledge translation and exchange efforts will be a central component of our goals and will be conducted using several approaches to reach the widest audience. First, at the end of the study, we will hold an end-of-grant virtual workshop and multiple presentations for all interested diverse stakeholders (e.g., study participants, HCPs, newcomer serving agencies, ethnocultural groups, officials from the NB Ministry of Health) to disseminate our findings, gather feedback, and address questions. Second, using our subscription to Canva (i.e., an infographic maker), we will create a series of infographics-based results and experience conducting this study. Infographics will also be created for dissemination across social media platforms. Third, to reach the academic and clinical community, we will present our findings at local, provincial, national, and/or international conferences and publish the results in peer-reviewed journals. We aim to publish in at least one open-access peer-reviewed journal article to increase the reach of our findings. Finally, we will submit a written final project report to the Assistant Dean of Dalhousie Medicine NB Research and to the Government of NB's Office of the Chief Medical Officer of Health (Public Health). A copy of the findings will also be shared with participants who desire to receive one.

## Supporting information

**S1 Appendix. Informed consent form in multiple languages.**
(PDF)

**S2 Appendix. Participant surveys multiple languages.**
(PDF)

**S3 Appendix. LTBI indicator form.**
(PDF)

**S4 Appendix. Health care practitioner survey.**
(PDF)

## Acknowledgments

We would like to acknowledge Dr. Andrew Flewelling for assistance with sample size calculation and Clara Kelly for help with participant recruitment.

## Author Contributions

**Conceptualization:** Isdore Chola Shamputa, Duyen Thi Kim Nguyen, Kimberley Barker, Duncan Webster.

**Data curation:** Isdore Chola Shamputa.

**Formal analysis:** Isdore Chola Shamputa, Duyen Thi Kim Nguyen.

**Funding acquisition:** Isdore Chola Shamputa, Duyen Thi Kim Nguyen, Kimberley Barker, Duncan Webster.

**Investigation:** Isdore Chola Shamputa, Duyen Thi Kim Nguyen, Doaa Higazy, Amani Abdelhadi, Duncan Webster.

**Methodology:** Isdore Chola Shamputa, Duyen Thi Kim Nguyen.

**Project administration:** Isdore Chola Shamputa, Duyen Thi Kim Nguyen, Michelle Reddin.

**Resources:** Isdore Chola Shamputa.

**Software:** Isdore Chola Shamputa, Duyen Thi Kim Nguyen.

**Supervision:** Isdore Chola Shamputa, Duyen Thi Kim Nguyen, Hope MacKenzie, Kimberley Barker, Duncan Webster.

**Validation:** Isdore Chola Shamputa, Duyen Thi Kim Nguyen, Duncan Webster.

**Visualization:** Isdore Chola Shamputa, Duyen Thi Kim Nguyen.

**Writing – original draft:** Isdore Chola Shamputa, Duyen Thi Kim Nguyen.

**Writing – review & editing:** Isdore Chola Shamputa, Duyen Thi Kim Nguyen, Doaa Higazy, Amani Abdelhadi, Hope MacKenzie, Michelle Reddin, Kimberley Barker, Duncan Webster.

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
