## [Decision Letter · Decision Letter 0]

11 Aug 2022

PONE-D-22-13477Optimizing tuberculosis screening for immigrants in southern New Brunswick: A pilot study ProtocolPLOS ONE

Dear Dr. Shamputa,

Thank you for submitting your manuscript to PLOS ONE. After careful consideration, we feel that it has merit but does not fully meet PLOS ONE’s publication criteria as it currently stands. Therefore, we invite you to submit a revised version of the manuscript that addresses the points raised during the review process.

We look forward to receiving your revised manuscript.

Kind regards,

Jinsoo Min

Academic Editor

PLOS ONE

Journal Requirements:

Additional Editor Comments:

Thank you for submitting the manuscript. Please, find the reviewers' comments and revised the original manuscript accordingly.

In addition, read our study protocol submission guideline for more information and follow the instructions.

Reviewers' comments:

Reviewer's Responses to Questions

**Comments to the Author**

1. Does the manuscript provide a valid rationale for the proposed study, with clearly identified and justified research questions?

Reviewer #1: Yes

Reviewer #2: Yes

2. Is the protocol technically sound and planned in a manner that will lead to a meaningful outcome and allow testing the stated hypotheses?

Reviewer #1: Yes

Reviewer #2: Partly

3. Is the methodology feasible and described in sufficient detail to allow the work to be replicable?

Reviewer #1: Yes

Reviewer #2: No

4. Have the authors described where all data underlying the findings will be made available when the study is complete?

Reviewer #1: Yes

Reviewer #2: Yes

5. Is the manuscript presented in an intelligible fashion and written in standard English?

Reviewer #1: Yes

Reviewer #2: Yes

6. Review Comments to the Author

You may also provide optional suggestions and comments to authors that they might find helpful in planning their study.

Reviewer #1: The protocol was well detailed but only deficient in the patient consent section. This is not detailed and considering the need to have a full disclosure the clients the consent form must be very detailed as such information as ; details of the study, purpose, procedures, risks or discomfort, costs to participants and confidentiality were not well presented

Reviewer #2: Thank you for the opportunity to review this manuscript. This study protocol describes a plan to for a pilot study to systematically screen newcomers from TB endemic countries for latent tuberculosis infection, using IGRA tests. My comments are provided to help strengthen the protocol and this manuscript.

Abstract – the abstract includes only the Introduction, objectives and methods sections, but would be better to summarize all sections of the manuscript, including the analysis plan, discussion, etc.

Recruitment – over 19 years of age – please explain why recruitment does not include adults aged 18 years

Delighted to see the intent to apply a SGBA approach. When referring to gender, in the participant recruitment (line 113), for example, will this be a demographic question about self-identity? In the demographics questionnaire, how will “culture” be defined, in comparison to ethnicity for example or country of origin? Similarly, in the participant survey (Likert scales) how will “cultural issues” be presented to respondents as something to comment on?

HCP survey – Please spell out what is meant by HCP when it is first used. What demographic information will be collected from providers? It will be important to consider gender and sex among this study population as well as the newcomer population. How many HCPs will be recruited to the study? Was the survey piloted for HCP also written at a grade 3 level?

Section 4 LTBI screening – some confounding of screening out participants for their eligibility in the study and screening for LTBI. This should be revised for clarity

LTBI follow- up – this portion will benefit from additional details about how stigma and treatment-related concerns will be addressed. Is this to be accomplished by this study team?

Participant and HCP surveys will be administered after LTBI follow-up – what is that timeframe? Does that refer to completion of treatment? Needs clarification.

Statistical Analyses – “predictors of the logistic regression” will be age, gender, country of origin, etc. I recommend using sex as one of the predictors. The progression of TB and other diseases is entwined with sex – anatomy and physiology, whereas gender – which refers to social norms, roles and identities, will also be important in terms of ability to participate in the study or in follow-up, but for different (social) reason.

Description of qualitative analyses is too brief. Will a constant comparison method be used for the analyses, for example? And how many research team members will be looking for and validating codes and themes?

Discussion: I recommend some mention of the benefit of reducing the burden of TB disease and suffering, not just as a health care cost saving.

7. PLOS authors have the option to publish the peer review history of their article (what does this mean?). If published, this will include your full peer review and any attached files.

Reviewer #1: No

Reviewer #2: **Yes: **M J Haworth-Brockman

---

## [Author Response · Author response to Decision Letter 0]

26 Aug 2022

Responses to Reviewer’s comments

Reviewer #1: 

The protocol was well detailed but only deficient in the patient consent section. This is not detailed and considering the need to have a full disclosure the clients the consent form must be very detailed as such information as ; details of the study, purpose, procedures, risks or discomfort, costs to participants and confidentiality were not well presented.

Authors’ response: The information requested is included on the consent form [see a copy of the English version below] [see also S1 Appendix]:

+++++++++++++++++++++++++++++++++++++++++++++++++++++++++++++++++++++++++++++++++++++++++++++

Informed Consent Form for Participants

Title of Project: Optimizing Tuberculosis Screening for Newcomers in Southern New Brunswick: A Pilot Study

Introduction

We are inviting adults 19 years of age and older to participate in a study on screening for latent tuberculosis infection. The purpose of this study is to gather information to assess the feasibility of implementing a latent tuberculosis infection screening program for newcomers from countries with many cases of tuberculosis.

If you choose to participate, you will be asked to provide about 4 ml of blood for testing latent tuberculosis infection and complete an online survey for approximately 10 minutes about your experience in participating in the study.

Your participation is voluntary, and you may withdraw from the study at any time without penalty. There is no cost for participating in the study. Participants will receive up to $40 in e-gift cards for their time, i.e., $20 after blood collection and another $20 after completing the online survey. 

Potential benefits

The potential benefits from participating in this study may include the reassurance that you do not have latent tuberculosis infection (if the test is negative). If the test is positive and you get treated, you will benefit by considerably reducing the chances of suffering from tuberculosis disease, thereby keeping your family and friends safe. 

Potential risks:

There is a minimal risk of discomfort, bruising, or infection at that site during blood collection, similar to giving an amount of blood for other medical tests. 

Privacy and confidentiality

Protecting your privacy is an important part of this study. Information gathered from this study is strictly confidential, and your information will be anonymized. All electronic data will be stored on a secure drive at the Department of Nursing & Health Sciences at the University of New Brunswick for 5 years to allow for the dissemination of information, after which it will be destroyed. The data will only be accessed by members of the study team. 

QUESTIONS 

If you have questions after you read this form, ask the research team member assessing you. You should not sign this form or provide verbal consent until you are sure that you understand the study. 

This project has been reviewed by the Research Ethics Board of the University of New Brunswick and is on file as REB File #033-2021, Brunswick, and the Horizon Health Network Research Ethics Board and poses minimal risk. In the event that injury, illness or disability results and you believe that it is related to your participation in this study, or if you have any questions about your rights as a research participant, you may contact Dr. Beth Keyes, Chair of the Research Ethics Board at the UNBSJ by phone [506-648-5994] or by email [REB@unb.ca] or Regional Director of Ethics Services, Horizon Health Network Research Ethics Board by phone [506) 648-6094] or by email at [REBOffice@HorizonNB.ca]. 

PARTICIPANTS STATEMENT 

I have read the information about this study and have had the opportunity to discuss this study and my questions have been answered to my satisfaction. I acknowledge that I have been informed that my participation is voluntary and that the data I provide will remain confidential. I hereby consent to take part in this study. 

Name of participant 

Signature of participant____________________________ Date: ______________

OR 

Obtained verbal consent of participant_________________ Date: ______________

Please note that by consenting to participating in this study, you have not waived any rights to legal recourse in the event of research-related harm. 

STATEMENT BY PERSON PROVIDING INFORMATION ON STUDY 

I have explained to the above participant the nature, requirements and the purpose of the study, potential benefits, and possible risks associated with participation in this study. I have answered any questions that have been raised. I believe that the participant understands the implications and the voluntary nature of the study. 

Researcher Signature: ____________________________ Date: ______________ 

Research Team:

Dr. Duncan Webster, Division of Infectious Diseases, Department of Medicine, Saint John Regional Hospital, duncan.webster@horizonnb.ca

Dr. Isdore Chola Shamputa, Department of Nursing & Health Sciences, University of New Brunswick Saint John, chola.shamputa@unb.ca

Dr. Kimberly Barker, Medical Officer of Health, South Region, New Brunswick Department of Health, Kimberley.Barker@gnb.ca

Dr. Duyen Nguyen, Regional Senior Program Advisor, Government of New Brunswick, duyen.nguyen@gnb.ca

++++++++++++++++++++++++++++++++++++++++++++++++++++++++++++++++++++++++++

Reviewer #2: 

Thank you for the opportunity to review this manuscript. This study protocol describes a plan to for a pilot study to systematically screen newcomers from TB endemic countries for latent tuberculosis infection, using IGRA tests. My comments are provided to help strengthen the protocol and this manuscript.

Abstract – the abstract includes only the Introduction, objectives and methods sections, but would be better to summarize all sections of the manuscript, including the analysis plan, discussion, etc.

Authors’ Response: The following information has been added to the Abstract to address this concern:

”For the quantitative portion of the study, descriptive statistics will be used to summarize participant characteristics and feedback. Simultaneous logistic regression will be performed to identify variables associated with the IGRA test outcome and evidence of increased CD8 T-cell immune response among those found to be LTBI-positive. Qualitative results will be analyzed using inductive thematic analysis.

Discussion: The findings from this study will allow us to understand the role of the IGRA LTBI screening assay and its feasibility and acceptability by immigrants and HCPs in New Brunswick (NB). The findings will additionally provide information on the enhancers and barriers of LTBI screening and management useful in determining how best to expand the latent TB screening program if deemed appropriate.” [see lines 43-52]. 

Recruitment – over 19 years of age – please explain why recruitment does not include adults aged 18 years

Authors’ response: Individuals below 19 years are considered minors in Canadian Province of New Brunswick and thus cannot legally provide consent to participate in the study; that is why we are including individuals who are at least 19 years old (see definition of majority age: https://www.canada.ca/en/immigration-refugees-citizenship/corporate/publications-manuals/operational-bulletins-manuals/refugee-protection/canada/processing-provincial-definitions-minor.html).

Delighted to see the intent to apply a SGBA approach. When referring to gender, in the participant recruitment (line 113), for example, will this be a demographic question about self-identity? 

Authors’ response: Yes. Gender is included as a demographic question [see enclosed LTBI Form below (S3 Appendix)].

~~~~~~~~~~~~~~~~~~~~~~~~~~~~~~~~~~~~~~~~~~~~~~~~~~~~~~~~~~~~~~~~~~~~~~~~~~

LTBI Indicator Form

Patient ID (assigned by researcher):____________________________

Patient Demographic Information:

1. Date of birth________________________ 

2. Sex: 1. Male 2. Female 

3. Gender: 1. Male 2. Female 3. Other: ________________

4. Country/Region of birth: ________________

5. Countries where you lived for more than 6 months (before coming to Canada):_________ ________________________________________________________________________________________________________________________________________________________________________________________________________________________

6. Date of immigration/arrival to Canada: _____________________

7. Immigration stream: 

1. Economic 2. Family Reunification 3. Study Permit

4. Work Permit 5. Refugee/Refugee Claimant 6. Temporary Visitors

Pre-Visit Data: (to be completed by Public Health)

8. New LTBI case? 1. Yes 2. No

9. Known prior contact with active TB? 1.Yes 2. No 3. Unknown; If yes, when?________

10. Client enrolled in IGRA study? 1. Yes 2. No

11. Medical Surveillance client?. 1. Yes 2. No

12. TST…………………………………….__________________(mm of induration)

13. IGRA……………………………………1. Positive 2. Negative 3. Not performed

14. IGRA result data…………….. TB1:________; TB2:_________; NIL:__________

15. Date of referral for LTBI treatment assessment:______________________________

Visit Data: (to be completed by assessing healthcare provider)

16. Referred LTBI patient seen within 30 days for treatment assessment. 1. Yes 2. No

17. Referred LTBI Patient seen within 90 days for treatment assessment. 

1. Yes 2. No 3. N/A

18. Referral source____________________Public Health / ID / Other: ________________

19. BCG vaccine 1. Yes 2. No 3. Unknown

20. Presence of comorbidities increasing risk for activation. 1. Yes 2. No

21. TST in 3D score……….% risk for active TB within next 2 years / % risk for active TB before age 80 years

22. Treatment 1. Yes 2. Prior adequate treatment 3. Refused

23. Planned treatment start date: ______________________________________________

24. Drug regimen chosen or previously administered: _____________________________

Post-Visit Data: (to be completed by assessing healthcare provider)

25. Completed treatment: 1. Yes 2. No

26. # missed doses: ________________________________________________________

27. Adverse effects reported: 1. Yes 2. No

Clinically significant ALT rise (>2-3x upper limit normal?). 1. Yes 2. No

28. 80% or greater compliance with risk adjusted follow up schedule: 1. Yes 2. No

29. Agrees to annual follow up CXR x 2 years. 1. Yes 2. No 3. N/A

~~~~~~~~~~~~~~~~~~~~~~~~~~~~~~~~~~~~~~~~~~~~~~~~~~~~~~~~~~~~~~~~~~~~~~~~~~~~~~

In the demographics questionnaire, how will “culture” be defined, in comparison to ethnicity for example or country of origin? 

Authors response: We thank the reviewer for bringing this point to our attention. Culture, was not explicitly defined and there are no direct questions on it in the demographic questionnaire. For this reason, culture has been removed from that statement.

Similarly, in the participant survey (Likert scales) how will “cultural issues” be presented to respondents as something to comment on?

Authors response: As stated above we do not have specific questions on culture, rather, there are questions such as questions 12, 13, 15 and 16 with open ended questions where participants can elaborate on their response to respective questions. This is where we hope to gain some insights into plausible cultural stigma. Below is the English version of the participant survey depicting the questions alluded to above [see also S2 Appendix]:

~~~~~~~~~~~~~~~~~~~~~~~~~~~~~~~~~~~~~~~~~~~~~~~~~~~~~~~~~~~~~~~~~~~~~~~~~~~~~~

Participant Survey

Instructions: Please circle the response that best describes your agreement with the statement. 

1. I received information on why the latent tuberculosis test was being done. 

Strongly Disagree Disagree Neutral Agree Strongly Agree

1 2 3 4 5

2. The blood collection site was easy to access.

Strongly Disagree Disagree Neutral Agree Strongly Agree

1 2 3 4 5

3. The blood collection site was easy to commute to. 

Strongly Disagree Disagree Neutral Agree Strongly Agree

1 2 3 4 5

4. The blood collection process was simple (e.g. registration, blood collection). 

Strongly Disagree Disagree Neutral Agree Strongly Agree

1 2 3 4 5

5. How long did the blood collection process take (in minutes)___________________________

6. The waiting time for blood collection was reasonable.

Too long Somewhat long Neutral Somewhat short Very short

1 2 3 4 5

7. I received adequate information regarding my latent tuberculosis test results.

Strongly Disagree Disagree Neutral Agree Strongly Agree

1 2 3 4 5

8. I was satisfied with the way my latent tuberculosis test result was communicated to me.

Strongly Disagree Disagree Neutral Agree Strongly Agree

1 2 3 4 5

9. The healthcare provider (i.e., doctor, nurse) clearly explained the test results.

Strongly Disagree Disagree Neutral Agree Strongly Agree

1 2 3 4 5

10. The healthcare provider answered all my questions well.

Strongly Disagree Disagree Neutral Agree Strongly Agree

1 2 3 4 5

11. I was treated with respect by the healthcare provider.

Strongly Disagree Disagree Neutral Agree Strongly Agree

1 2 3 4 5

12. I was satisfied with the overall experience with the latent tuberculosis screening process and/or care I received. 

Strongly Disagree Disagree Neutral Agree Strongly Agree

1 2 3 4 5

13. I did not experience any barriers when participating in this study.

Strongly Disagree Disagree Neutral Agree Strongly Agree

1 2 3 4 5

14. I would recommend latent tuberculosis screening services to other newcomers.

Strongly Disagree Disagree Neutral Agree Strongly Agree

1 2 3 4 5

15. I did not feel stigmatized when participating in the study.

Strongly Disagree Disagree Neutral Agree Strongly Agree

1 2 3 4 5

16. I had concerns about participating in the study.

Strongly Disagree Disagree Neutral Agree Strongly Agree

1 2 3 4 5

17. My knowledge, attitudes, beliefs, and/or fear regarding tuberculosis improved by participating in the study.

Strongly Disagree Disagree Neutral Agree Strongly Agree

1 2 3 4 5

18. Do you have any suggestions on how the latent tuberculosis screening process can be improved?

 1. Yes 2. No

~~~~~~~~~~~~~~~~~~~~~~~~~~~~~~~~~~~~~~~~~~~~~~~~~~~~~~~~~~~~~~~~~~~~~~~~~~~~~~

HCP survey – Please spell out what is meant by HCP when it is first used. 

Authors’ response: Health Care Provider (HCP) has been spelt out on its first use [see line 36].

What demographic information will be collected from providers? 

Authors’ response: Sex and gender. 

It will be important to consider gender and sex among this study population as well as the newcomer population. 

Authors’ response: We can confirm that both sex and gender were considered for both study populations.

How many HCPs will be recruited to the study? 

Authors’ response: There are two health care providers on the research team that will care for all study participants; an infectious diseases physician and a Nurse Practitioner. We have an option of adding another Nurse Practitioner if the need arises.

Was the survey piloted for HCP also written at a grade 3 level?

Authors’ response: Yes it was.

Section 4 LTBI screening – some confounding of screening out participants for their eligibility in the study and screening for LTBI. This should be revised for clarity

Authors’ response: The inclusion criteria targets participants with a high risk for latent TB infection. These are individuals who can benefit most from the study and from whom we can gather most useful information regarding how best the latent TB screening can be utilized, if used widely in New Brunswick. For the type of population and location we are dealing with, we feel that our current criteria is robust enough to identify the most relevant participants. The potential for confusion is limited and can easily be addressed if in doubt because only a few members of the research staff dealing with recruitment are very abreast with the criteria. It is possible that the target population in other settings may be slightly different.

LTBI follow- up – this portion will benefit from additional details about how stigma and treatment-related concerns will be addressed. Is this to be accomplished by this study team?

Authors’ response: TB stigma and treatment concerns are being partially addressed in this study but are also being considered as a separate undertaking by the research team. In this study, discussions regarding TB-related stigma and treatment concerns commence during recruitment. In addition, when delivering the participants' latent TB test results, the team's infectious diseases physician addresses these concerns. The infectious diseases physician also assesses, counsels, and may offer treatment to those with positive test results, if indicated. This channel of communication will remain open to the participants during treatment follow up. A Nurse Practitioner is also available an as extra resource. In addition, there are separate ongoing efforts by the research team focused on addressing TB- stigma-related concerns and providing education on TB, particularly the differences between active and latent tuberculosis and the associated risks. For example, the study team speaks to different ethnocultural groupings about TB, answering questions and dispelling misinformation in the community. The team has also developed various leaflets regarding TB as a follow up to similar findings from our previous study with a similar population (Shamputa et al., 2022).

Participant and HCP surveys will be administered after LTBI follow-up – what is that timeframe? Does that refer to completion of treatment? Needs clarification.

Authors’ response: Participant questionnaires administration will be done as the last step of participating in the study. For participants with a negative latent TB result, the survey will be administered soon after the physician informs them of their test results. Typically, this is within 2 weeks of providing a sample for testing. For participants with a positive latent TB result, the survey is administered after the participant completes treatment about 4-months later (if they accepted treatment) or soon after they’ve had a discussion with the physician about their test results, if they decline treatment. The following statement has been added to the manuscript 

“ i i) the participant has received a negative IGRA result, ii) after the completion of treatment for those with a positive IGRA result, were assessed by a HCP, offered and agreed to be treated, or iii) after receiving a positive IGRA result, were assessed by a HCP, offered but declined treatment.” [see lines 218-221]. HCP surveys will be administered at the end of the study [see lines 222-223].

Statistical Analyses – “predictors of the logistic regression” will be age, gender, country of origin, etc. I recommend using sex as one of the predictors. The progression of TB and other diseases is entwined with sex – anatomy and physiology, whereas gender – which refers to social norms, roles and identities, will also be important in terms of ability to participate in the study or in follow-up, but for different (social) reason.

Authors’ response: As per Reviewer 2’s recommendation, sex has been added as one of the predictors [see line 236] (six in total now). Note that this has resulted in the increase in the sample size to 288 (and updated throughout the manuscript).

Description of qualitative analyses is too brief. Will a constant comparison method be used for the analyses, for example? And how many research team members will be looking for and validating codes and themes?

Authors’ response: This section has been expanded in response to this observation and a reference regarding the “constant” method that will be used to conduct thematic analysis added. That section now reads as follows 

“...as described elsewhere [25]. Briefly, two research team members will independently familiarize themselves with all of the data. Next, initial codes will be generated independently and collaboratively with a third-team member. Subsequently, codes will be integrated into emerging themes; these will be reviewed by the entire team and modified if necessary” [see lines 249-252 and 410-411]. 

Three research members with expertise in qualitative data analyses will perform the primary analysis, which will be shared and discussed with the entire research team. 

Discussion: I recommend some mention of the benefit of reducing the burden of TB disease and suffering, not just as a health care cost saving.

Authors’ response: As recommended by reviewer 2, the following information has been added to the manuscript: 

“Further, the detection and treatment of individuals with latent TB have several other advantages; first, given that an estimated one-quarter of the world's human population is infected with TB [1], detecting and treating latent TB is one of the critical strategies for the elimination of TB as a global public health threat [26]. Second, mathematical modeling studies have demonstrated that through screening and control strategies targeting latent TB, the development of active TB can be reduced between 20.6 [27] and 40% on a population level [28]. Third, the prevention of active TB by detecting and treating latent TB can also help prevent numerous health problems and post-TB sequelae that may be experienced by individuals despite adequate and successful treatment of active TB [29].” [see lines 280-288] and the respective citations have been added to the reference list [see lines 412-422].

---

## [Decision Letter · Decision Letter 1]

13 Oct 2022

PONE-D-22-13477R1Optimizing tuberculosis screening for immigrants in southern New Brunswick: A pilot study ProtocolPLOS ONE

Dear Dr. Shamputa,

Thank you for submitting your manuscript to PLOS ONE. After careful consideration, we feel that it has merit but does not fully meet PLOS ONE’s publication criteria as it currently stands. Therefore, we invite you to submit a revised version of the manuscript that addresses the points raised during the review process. Please submit your revised manuscript by Nov 27 2022 11:59PM. If you will need more time than this to complete your revisions, please reply to this message or contact the journal office at plosone@plos.org. Please include the following items when submitting your revised manuscript:A rebuttal letter that responds to each point raised by the academic editor and reviewer(s). You should upload this letter as a separate file labeled 'Response to Reviewers'.A marked-up copy of your manuscript that highlights changes made to the original version. You should upload this as a separate file labeled 'Revised Manuscript with Track Changes'.An unmarked version of your revised paper without tracked changes. You should upload this as a separate file labeled 'Manuscript'.If applicable, we recommend that you deposit your laboratory protocols in protocols.io to enhance the reproducibility of your results. Protocols.io assigns your protocol its own identifier (DOI) so that it can be cited independently in the future. For instructions see: https://journals.plos.org/plosone/s/submission-guidelines#loc-laboratory-protocols. Additionally, PLOS ONE offers an option for publishing peer-reviewed Lab Protocol articles, which describe protocols hosted on protocols.io. Read more information on sharing protocols at https://plos.org/protocols?utm_medium=editorial-email&utm_source=authorletters&utm_campaign=protocols.

We look forward to receiving your revised manuscript.

Kind regards,

Jinsoo Min

Academic Editor

PLOS ONE

Journal Requirements:

Additional Editor Comments (if provided):

(1) Please, define the abbreviations if it used first in the manuscript body. (HCP, NB, PC)

For example, full name of 'HCP' is missing in both abstract and manuscript body.

(2) Please, minimize the use of the abbreviation.

For exmaple, I think that it is unecessary to use "NB" in the abstract, because it is used only one time in the abstract. Another example is "PC".

(3) I think you need to insert branckets before and after "SGBA+".

(4) Please, check any minor erros. PLOS ONE does not copyedit accepted manuscripts.

Reviewers' comments:

Reviewer's Responses to Questions

**Comments to the Author**

1. Does the manuscript provide a valid rationale for the proposed study, with clearly identified and justified research questions?

Reviewer #2: Yes

2. Is the protocol technically sound and planned in a manner that will lead to a meaningful outcome and allow testing the stated hypotheses?

Reviewer #2: Yes

3. Is the methodology feasible and described in sufficient detail to allow the work to be replicable?

Reviewer #2: Yes

4. Have the authors described where all data underlying the findings will be made available when the study is complete?

Reviewer #2: Yes

5. Is the manuscript presented in an intelligible fashion and written in standard English?

Reviewer #2: Yes

6. Review Comments to the Author

You may also provide optional suggestions and comments to authors that they might find helpful in planning their study.

Reviewer #2: I am satisfied that the authors have made the necessary revisions to this manuscript and it is ready for publication. I note that the authors comment that data will be presented in their final form in aggregated form, however I encourage the authors to continue with plans to analyze AND present the data by sub-tabulations as possible (e.g., by sex) so that the results can be most useful to other researchers and LTBI program managers.

7. PLOS authors have the option to publish the peer review history of their article (what does this mean?). If published, this will include your full peer review and any attached files.

Reviewer #2: No

---

## [Author Response · Author response to Decision Letter 1]

13 Oct 2022

Responses to Reviewers’ comments:

Below are our responses to the reviewers’ comments: 

(1) Please, define the abbreviations if it used first in the manuscript body. (HCP, NB, PC)

For example, full name of 'HCP' is missing in both abstract and manuscript body.

Response: i) “HCP” has been defined in line 36; ii) NB and PC are addressed below:

(2) Please, minimize the use of the abbreviation.

For example, I think that it is unnecessary to use "NB" in the abstract, because it is used only one time in the abstract. Another example is "PC".

Response: i) “NB” abbreviation has been deleted from the abstract and introduced in the manuscript body (see line 81) and ii)“PC”, this abbreviation has been deleted and replaced by “Personal Computers.” (see lines 257 and 258).

(3) I think you need to insert brackets before and after "SGBA+".

Response: brackets have been inserted (see line 233).

(4) Please, check any minor errors. PLOS ONE does not copyedit accepted manuscripts.

Response: This has been done.

Reviewer #2, “...I note that the authors comment that data will be presented in their final form in aggregated form, however I encourage the authors to continue with plans to analyze AND present the data by sub-tabulations as possible (e.g., by sex) so that the results can be most useful to other researchers and LTBI program managers.”

Response: This will be done.

---

## [Editor Report · Decision Letter 2]

17 Oct 2022

PONE-D-22-13477R2Optimizing tuberculosis screening for immigrants in southern New Brunswick: A pilot study ProtocolPLOS ONE

Dear Dr. Shamputa,

Thank you for submitting your manuscript to PLOS ONE. After careful consideration, we feel that it has merit but does not fully meet PLOS ONE’s publication criteria as it currently stands. Therefore, we invite you to submit a revised version of the manuscript that addresses the points raised during the review process.

We look forward to receiving your revised manuscript.

Kind regards,

Jinsoo Min

Academic Editor

PLOS ONE

Journal Requirements:

Additional Editor Comments:

(1) Define abbreviations upon first appearance in the text.

: IGRA is not defined in the manuscript body.

(2) Keep abbreviations to a minimum.

: 'COVID-19' is only used once. Please, change the "COVID-19" to "coronavirus disease 2019".

(3) The author states that "This sutdy uses the fourth-generaion IGRA". I thick this assays is the QuantiFERON-TB Gold Plus. Please, mention the name of the assay and its manufacturer in an appropirate place.

(4) Please, check any minor erros. PLOS ONE does not copyedit accepted manuscripts.

---

## [Author Response · Author response to Decision Letter 2]

18 Oct 2022

Editor’s comment:

(1) Define abbreviations upon first appearance in the text.

IGRA is not defined in the manuscript body.

Authors’ Response: 

IGRA has been defined in the manuscript (see line 94).

Editor’s comment:

(2) Keep abbreviations to a minimum.

'COVID-19' is only used once. Please, change the "COVID-19" to "coronavirus disease 2019".

Authors’ Response: 

As suggested, COVID-19" has now been changed to "coronavirus disease 2019" (see line 194).

Editor’s comment:

(3) The author states that "This study uses the fourth-generaion IGRA". I thick this assays is the QuantiFERON-TB Gold Plus. Please, mention the name of the assay and its manufacturer in an appropriate place.

Authors’ Response:

That is correct. The statement in question has been updated and now reads as follows, 

“This study is using the QuantiFERON-TB Gold Plus (QFT-Plus) interferon-gamma release assay (IGRA) (QIAGEN),....” (see lines 93-94).

Editor’s comment:

(4) Please, check any minor errors. PLOS ONE does not copyedit accepted manuscripts.

Authors’ Response: 

This has been done.

---

## [Editor Report · Decision Letter 3]

24 Oct 2022

Optimizing tuberculosis screening for immigrants in southern New Brunswick: A pilot study Protocol

PONE-D-22-13477R3

Dear Dr. Shamputa,

We’re pleased to inform you that your manuscript has been judged scientifically suitable for publication and will be formally accepted for publication once it meets all outstanding technical requirements.

Kind regards,

Jinsoo Min

Academic Editor

PLOS ONE